# Robot-enhanced diabetes care for middle-aged and older adults living with diabetes in the community: A small sample size mixed-method evaluation

**Ching-Ju Chiu**[1], **Lin-Chun Hua**[1]*, **Chieh-Ying Chou**[1,2], **Jung-Hsien Chiang**[3]

**1** Institute of Gerontology, College of Medicine, National Cheng Kung University, Tainan, Taiwan,
**2** Department of Family Medicine, National Cheng Kung University Hospital, College of Medicine, National Cheng Kung University, Tainan, Taiwan, **3** Department of Computer Science and Information Engineering, College of Electrical Engineering and Computer Science, National Cheng Kung University, Tainan, Taiwan

\* flower921flora@gmail.com

**Data Availability Statement:** All relevant data are within the paper.

## Abstract

### Purpose

This study assessed robot-enhanced healthcare in practical settings for the purpose of community diabetes care.

### Methods

A mixed method evaluation collected quantitative and qualitative data on diabetes patients over 45 (N = 30) and community pharmacists (N = 10). It took 15–20 min for the diabetes patients to interact with the robot. Before and after the interaction, questionnaires including a diabetes knowledge test, self-efficacy for diabetes, and feasibility of use of the robot was administered. In-depth interviews with both pharmacists and patients were also conducted.

### Results

After interacting with the robot, a statistically significant improvement in diabetes knowledge ($p < .001$) and feasibility of the robot ($p = .012$) was found, but self-efficacy ($p = .171$) was not significantly improved. Five themes emerged from interviewing the diabetes patients: Theme 1: meets the needs of self-directed learning for the elderly; Theme 2: reduces alertness and creates comfortable interaction; Theme 3: vividness and richness enhance interaction opportunities; Theme 4: Robots are not without disadvantages, and Theme 5: Every person has unique tastes. Three themes emerged from interviewing pharmacists: Theme 1: Technology must meet the real needs of the patient; Theme 2: creates new services, and Theme 3: The use of robots must conform to real-life situations.

### Conclusions

Both the diabetes patients and the pharmacist reported more positive feedback on the robot-enhanced diabetes care than concerns. Self-directed learning, comfortable

**Funding:** This work was supported by a grant from the Taiwan Ministry of Science and Technology (MOST 108-2634-F-006 -006- ). CJC received the award. The funder had no role in study design, data collection and analysis, decision to publish, or preparation of the manuscript.

**Competing interests:** The authors have declared that no competing interests exist.

**Abbreviations:** COPD, chronic obstructive pulmonary disease; DKT, Diabetes Knowledge Scale; ECG, electrocardiography; IADLs, instrumental activities of daily living; mHealth, mobile health; SES, Self-Efficacy Scale; TAM, Technology Acceptance Model.

interaction, and vividness were the most focuses when using robot to enhance self-management for the patients. Pharmacists were most receptive to fit conforming with reality and creating new services.

## Introduction

Disease management, another aspect of technological health, is already being supported by technology for a number of chronic diseases. Thus, according to the literature, technological interventions can help stabilize diabetes, lung disease, and cardiovascular disease [1]. Interventions include the use of smartphones to assist in the management of diabetes [2]; the use of mHealth in combination with blood pressure measurement devices, ECG, and other recorders to monitor the condition of patients with heart failure [3], and the use of mHealth to improve asthma control and quality of life [4]. However, it is important to note that disease control goals and monitoring values vary greatly by age and even by individual, and cannot be applied as a whole. Therefore, the severity of a disease may be overlooked if there is a lack of personalization, or if the functions are limited. For example, if remote monitoring is only done once a day, heart failure may not be detected immediately [3]. It is therefore important to consider the nature of the disease and the heterogeneity of the patients in order to make the use of technology in disease management more valuable.

Even though it is understood that technology can be beneficial to the elderly, it does not mean that all elderly people are willing to use it. In terms of attitudes and acceptance of technology, older people tend to be less motivated to use it due to lack of experience and deterioration in their physical functions. According to a survey conducted by the National Development Council, 95% of mobile phone users in Taiwan at or under the age of 50 have used mobile phones to access the Internet, but this percentage drops to 83% for those at or over the age of 50, and drops as low as 54.5% for those at or over the age of 60 [5]. Older people with no experience with using technology are more reluctant to change the status quo and are more fearful of learning about the unknown [6]. In addition, when it takes extra effort to learn something new, and they do not understand the benefits, they are more likely to be resistant to technological interventions [7]. However, a negative judgement of technology can be reversed after experimentation. In other words, the actual use of technological software can change the attitudes of older individuals who were originally resistant to it [6]. Therefore, the designs of technologies should focus on an underlying theory, the Technology Acceptance Model (TAM). The ease of use and practicality of technological products is a key factor in the acceptance of new technologies [8]. Therefore, in order to make technology accessible to the middle-aged and elderly, it is important to have a simple, age-friendly user interface to increase willingness to try it. It is also important to enhance the practicality of the technology, i.e. the intervention must be highly relevant to the needs of the user [7].

Current applications of robots in middle-aged and elderly healthcare include the use of companionship to enhance social interaction and to provide assistance with disease management and improve compliance with medical advice [9, 10]. Interaction with elderly people with dementia through pet-like machines can enhance psychological and even social functioning [11]. In the case of chronic obstructive pulmonary disease (COPD) patients, robots can be used to improve quality of life and medication compliance through disease management assistance [10]. Thus, technological advances can be used to optimize the health status of middle-aged and elderly individuals from a broader perspective, thereby improving their quality of life. However, there is a lack of documentary evidence on the involvement of health technology

in the health sector, and the focus in the health field is on the patient, with little inclusion of medical professionals. If the use of technology is not scrutinized carefully enough, it may not meet the patient's needs, be in line with medical practices, or may even contradict medical objectives. In addition, healthcare interventions need to be long-term, targeted, and systematic. The role of professionals is essential in this regard. Through an analysis and evaluation of medical staff, effective feedback can be given to patients, and good two-way feedback is established, which demonstrates the value of the overall intervention [12]. Therefore, the use of technology in healthcare should make it possible to effectively link the patient side, the technology side, and the medical side, in order to improve overall quality of care and make such changes sustainable.

The objectives of this study are as follows:

Objective 1: To assess the satisfaction, diabetes knowledge, self-management, and attitude changes in middle-aged and older diabetic patients after using the prototype robot and to obtain overall evaluations and recommendations related to its use.

Objective 2: To explore the overall attitudes toward and evaluations of community pharmacists on the use of the prototype robot for the purpose of community diabetes care.

## Method

### Participants

The study was conducted in community pharmacies in southern Taiwan (Greater Tainan area) for the purpose of obtaining participants. Because it was a prototype robot test, it would be a small sample size study. The participants in this study were divided into two categories, diabetic patients and pharmacists in community pharmacies. The case intake criteria for the middle-aged and elderly diabetic participants included the following: (1) 45 years of age or older, (2) type 2 diabetic, (3) receiving a slow writing device from a community pharmacy, (4) able to communicate in Mandarin, (5) able to express their ideas in the form of language, and (6) agreeing to sign the participant consent form. Exclusion criteria included: (1) those with severe hearing impairment, (2) those who rely on aids or wheelchairs to get around, and (3) those with cognitive impairment. Finally, after the robotic intervention, semi-structured qualitative interviews were conducted for those participants with good language skills ability to express their ideas in the self-assessment. The case intake criteria for the community pharmacist participants included the following: (1) a community pharmacist who has been practicing at a community pharmacy in Tainan for at least one year, (2) currently practicing at a health insurance contract pharmacy and not a chain pharmacy, and (3) agreeing to sign a consent form as a participant.

### Procedures

This study was performed in accordance with the relevant guidelines and regulations, including the Declaration of Helsinki and was approved by the Institution Review Board (IRB) of National Cheng Kung University Hospital in Taiwan (No. A-ER-105-509). The study was conducted by the investigator himself. Information about the case intake process and the study were provided during the recruitment process, after which the participants signed the study consent form after giving verbal consent. The case intake process was in accordance with Human Subjects Committee procedures. Participants with diabetes completed a pre-test before interacting with the robot and were guided and assisted by the investigator for

approximately 15–20 minutes. The interaction mainly involved completing the health education game and learning about other basic functions and applications. During the interaction, the pharmacist from the community pharmacy participated in the process or addressed questions from the patients as appropriate. The objective of the quantitative evaluation was to analyze the feasibility of using the robot and the effectiveness of this form of health education.

## Measures

There were three main types of measurement instruments used in this study: The first was a user self-assessment questionnaire; the second was a semi-structured interview protocol for diabetic patients, and the last was a semi-structured interview protocol for healthcare providers. The self-administered questionnaire contains items on demographic variables, health status, and technology use experience, as well as a diabetes knowledge questionnaire, a diabetes self-care behavior questionnaire, community pharmacy robot application feasibility questionnaire, and a satisfaction questionnaire. The first part of the scale is the Diabetes Knowledge Scale, which is a questionnaire used to test knowledge about diabetes. In this study, the Revised Michigan Diabetes Knowledge Scale (DKT) [13] was used (supported by Grant Number P30DK020572 (MDRC) from the National Institute of Diabetes and Digestive and Kidney Diseases). Due to the time and content constraints related to the actual interaction with the robot, the interactive teaching content only contained 10 questions [13, 14]. The second part of the lesson was intended to understand the self-care behavior of the diabetic patient and to assess the patient's confidence related to engaging in various activities, such as diabetes self-care. The Chinese translation of the Self-Efficacy Scale (SES) [15] was used for this assessment. The feasibility of using a community pharmacy robot was the third part of the scale. It focuses on knowledge [13], disease management [15], disease anxiety relief [16] and interaction with a community pharmacist, respectively, as subjectively perceived by the diabetic patient. It was developed from a five-point Likert scale. The satisfaction survey is a 10-question questionnaire based on the use and satisfaction theory [17], with questions categorized as process gratification, content gratification, social gratification, overall satisfaction, and information and quality [18]. The items were scored on a five-point Likert scale.

The interview protocol for diabetic patients consisted of three main sections: suggestions and expectations of the current technological development, needs for diabetes health care, and expectations for future applications and development of robots; the interview protocol for healthcare providers consisted of suggestions and expectations on the current technological development, needs for diabetes care, the practical needs of pharmacists in community pharmacies, and expectations for future applications and development of robots.

## Analysis

The quantitative results of this study were firstly analyzed using descriptive statistics on the demographic and basic variables of the participants, where the results were presented as percentages. The satisfaction results were presented as mean scores using descriptive statistics. Secondly, a paired sample t-test was used to analyze whether there was a significant difference in the changes in the pre-post test scores for the Diabetes Knowledge Scale, the Diabetes Self-Efficacy Scale, and the feasibility of using the Community Pharmacy Robot. Finally, the Mann-Whitney U-test was used to analyze whether there were differences among the variables in terms of gender, age, education, experience with technology, and length of diabetes history. These analyses were conducted using SPSS v17.0 statistical software.

The semi-structured qualitative interviews were conducted with the aid of an interview outline and were recorded throughout the interview with the consent of the participant and

supplemented by note taking. After the interview, the audio recording was converted into a verbatim transcript in order to match the analysis of the data as closely as possible to the facts presented by the interviewees. A semi-structured outline of the interview was used as a first step, and the verbatim transcript was read repeatedly to mark key points and analyze them paragraph by paragraph. The core themes were established through a discussion between the researcher and the expert and by consolidating the themes [19]. The reliability and validity of the qualitative interviews are cross-checked using triangulation of the data [20].

## Results

### Characteristics

A total of 30 diabetic patients and 10 pharmacists from community pharmacies were enrolled in this study. Table 1 presents the characteristics of the 30 diabetic participants, who were an average age of 69.2 years (age range 45–85 years). Of the 30 participants, 15 (50%) were male and 15 (50%) were female. Half of the participants had a history of diabetes for more than 10 years (n = 15, 50%), while 26.7% (n = 8) and 23.3% (n = 7) had a history of less than 5 years and 6–10 years, respectively. Almost half of the participants were unfamiliar with the use of technology, and 50.0% had no previous experience with the Internet (n = 15). In terms of experience with robots, the majority of participants were not new to the concept of robots, with most (n = 18, 60.0%) having heard of, seen, or used robots. A total of 12 diabetic patients participated in the semi-structured interviews (Table 1), who were an average age of 65.75 years, half of whom were female (n = 6, 50%), had a history of diabetes for more than 10 years (n = 6, 50%), and had a secondary to high school education (n = 6, 50%). A total of 13 community pharmacies were enrolled in the study, and 10 pharmacists were interviewed from 10 of these pharmacies. Of the 10 pharmacists (Table 2), most were male (n = 9, 90%), with an average age of 65.75, and half of them were under 45 years old (n = 5, 50%). In terms of years of experience, pharmacists had an average of 17.8 years of experience in community pharmacies (range: 1–35 years).

**Satisfaction with the use of robots and changes in knowledge, self-management, and attitudes towards diabetes before and after the intervention.** The satisfaction scale consisted of 10 questions and was subdivided into five sub-themes: process satisfaction, content satisfaction, social satisfaction, overall satisfaction, and information quality. The combined results (Table 3) show that the mean score for each question was greater than 3 out of 5 on the middle scale, indicating that diabetic participants were satisfied with their experience with interacting with the robots. The highest level of satisfaction was "interacting with the community pharmacy robot gives me the opportunity to reach out to other people with diabetes in the community or to share my experiences and opinions with others," with 93.3% (n = 28) agreeing or strongly agreeing and a mean score of 4.27 (SD = .691). The item with the lowest score was "Overall, the Community Pharmacy Board robot functions as I would expect," with only 66.7% (n = 20) have a positive judgement, with a mean score of less than 4 (3.80, SD = 1.157), and this was also the highest negative judgment (16.6%, n = 5).

The results of the pretest-posttest analysis of the three questions on diabetes knowledge, confidence in diabetes self-care behaviors and the feasibility of using a robot are presented in Table 4. After interacting with the robot, the post-test scores increased significantly for the knowledge of diabetes question and the feasibility of using the robot, respectively. For the diabetes knowledge section, the participant's raw pre-test knowledge score was 5.83 (SD = 2.167), and the post-test score improved to 7.03 (SD = 2.236), indicating a significant improvement in the participant's knowledge of diabetes through interaction with the robot on health education (t(29) = 4.466, $p$-value < 2.466). 4.466, $p$-value <0.001); in the applied feasibility questionnaire,

**Table 1. Characteristics of the diabetic participants.**

| Variable | Questionnaire section n(%) or mean±SD | Semi-structured interview n(%) or mean±SD |
|---|---|---|
| | N = 30 | N = 12 |
| **Demographic variables** | | |
| Sex | | |
| Male | 15(50.0) | 6(50.0) |
| Female | 15(50.0) | 6(50.0) |
| Age (range 45–85 years) | 69.2±9.690 | 65.75±9.640 |
| 45–64 | 8(26.7) | 5(41.7) |
| 65–74 | 13(43.3) | 4(33.3) |
| 75 years and over | 9(30.0) | 3(25) |
| Place of Residence | | |
| Urban area | 20(66.7) | |
| Countryside | 10(33.3) | |
| Education Level | | |
| Primary School and below | 11(36.7) | 2(16.7) |
| Secondary and High School | 13(43.3) | 6(50.0) |
| University or above | 6(20.0) | 4(33.3) |
| **Health status** | | |
| History of diabetes mellitus | | |
| Less than 5 years | 8(26.7) | 4(33.3) |
| 6–10 years | 7(23.3) | 2(16.7) |
| 10+ years | 15(50.0) | 6(50.0) |
| History of other diseases | | |
| Hypertension | 21(70.0) | 8(66.7) |
| Hyperlipidemia | 10(33.3) | 3(25) |
| Cardiac disease | 9(30.0) | 5(41.7) |
| Arthritis | 5(16.7) | 3(25) |
| Renal Disease | 2(6.7) | |
| Chronic Liver Disease | 1(3.3) | |
| **Technology experience** | | |
| Internet experience | | |
| No experience | 15(50.0) | 4(33.3) |
| Less than 5 years of experience | 5(16.7) | 3(25) |
| 6 years or more | 10(33.3) | 5(41.7) |
| Experience using mobile applications | | |
| No experience at all | 13(43.3) | 3(25) |
| Have used, but do not know how to download applications (apps) | 10(33.3) | 8(66.7) |
| Can download apps and operate them by oneself | 7(22.3) | 1(8.3) |
| Experience of using robots | | |
| Never heard of or used it at all | 12(40.0) | 5(41.7) |
| Have heard of or seen others use it but have not used it myself | 16(53.3) | 6(50.0) |
| Have experience with using robots | 2(6.7) | 1(8.3) |

**Table 2. Characteristics of pharmacist respondents (n = 10) [a].**

| Variable | n(%) or mean±SD |
|---|---|
| **Gender** | |
| Male | 9(90.0) |
| Female | 1(10.0) |
| **Age (range: 28–71 years)** | 49.6±14.057 |
| 28–45 | 5(50.0) |
| 46–64 | 4(40.0) |
| 65 years old and above | 1(10.0) |
| **Years of experience in the community pharmacy (range: 1–35 years)** | 17.80±13.57 |
| 1–10 years | 5(50.0) |
| 10–30 years | 1(10.0) |
| 30+ years | 4(40.0) |
| **Distribution of community pharmacies** | |
| Urban areas | 9(90.0) |
| Countryside | 1(10.0) |

[a] Urban areas are defined as urbanized areas: those with a population density of 300 or more people per square kilometers [21].

**Table 3. Summary of satisfaction ratings [a-b].**

| Questions on the Satisfaction Questionnaire | M(SD) | Negative judgement n (%) | Neutral judgement n (%) | Positive judgement n (%) |
|---|---|---|---|---|
| **Process satisfaction** | | | | |
| Pleasant | 4.00(.98) | 4(13.3) | 2(6.7) | 24(80.0) |
| Novelty of access to technology | 4.03(.96) | 3(10.0) | 4(13.3) | 23(76.7) |
| Smooth and easy | 4.00(1.05) | 4(13.3) | 1(3.3) | 25(83.3) |
| Average score | **4.0111 (.94)** | | | |
| **Content Satisfaction** | | | | |
| Getting more information about diabetes | 4.10(.96) | 3(10.0) | 3(10.0) | 24(80.0) |
| Can share with friends and family | 4.07(.91) | 3(10.0) | 2(6.7) | 25(83.3) |
| Average score | **4.0833 (.92)** | | | |
| **Social Satisfaction** | | | | |
| Facilitated my communication with pharmacists in the community | 3.97(.93) | 2(6.7) | 7(23.3) | 21(70.0) |
| Having the opportunity to meet people with diabetes and share experiences | 4.27(.69) | 1(3.3) | 1(3.3) | 28(93.3) |
| Average score | **4.1167 (.75)** | | | |
| **Overall Satisfaction** | | | | |
| Functionality meets expectations | 3.80(1.16) | 5(16.6) | 5(16.7) | 20(66.7) |
| Average score | **3.80(1.16)** | | | |
| **Quality of information** | | | | |
| The information is reliable. | 4.07(.69) | 0 | 6(25.0) | 24(75.0) |
| I can clearly understand the message being delivered. | 4.17(.75) | 1(3.3) | 3(10.0) | 26(86.7) |
| Average score | **4.1167 (.67)** | | | |

**Table 4. Summary of paired t-tests for diabetic patients on pretest-posttest changes in interactions with health care robots [a-d].**

| Scale | Pre-test M(SD) | Post-test M(SD) | Degree of freedom (df) | t value (t) | p value (p) |
|---|---|---|---|---|---|
| Diabetes Knowledge Questions | 5.83 (2.17) | 7.03 (2.24) | 29 | 4.466 | .000** |
| Diabetes Self-Care Behavioral Confidence | 7.55 (1.56) | 7.96 (1.78) | 29 | 1.403 | .171 |
| Application feasibility | 14.10 (3.9) | 16.13 (3.48) | 29 | 3.287 | .003* |
| Increase health knowledge of the disease | 3.43 (1.01) 1–5 | 4.1 (.80) 2–5 | 29 | 3.673 | .001* |
| Help with disease management | 3.63 (.85) 1–5 | 4.00 (1.02) 1–5 | 29 | 1.884 | .070 |
| Reduce anxiety about the disease | 3.47 (1.04) 1–5 | 4.07 (.91) 1–5 | 29 | 3.168 | .004* |
| Facilitate interaction with pharmacists in community pharmacies | 3.57 (1.01) 1–5 | 3.97 (1.00) 1–5 | 29 | 2.350 | .026 |

[a] * $p < 0.05$

[b] ** $< 0.001$

[c] Total score of 10 for the question on diabetes knowledge; average score for confidence in diabetes self-care behaviors; total score of 20 for feasibility of use of the application

[d] strongly agree = 5, agree = 4, no opinion = 3, disagree = 2, strongly disagree = 1.

the mean total score for the pre-test was 14.10 (SD = 3.9), and the post-test score increased significantly ($p$ = .003) to 16.13 (SD = 3.48).

There were four sub-questions in the application feasibility measurement questionnaire. The post-test score change was significant for the first item "interaction with the community pharmacy robot helped me to increase my knowledge of health-related illnesses" ($p$ = .001), but was the lowest on the pre-test (3.43 ± 1.01). However, the post-test score not only increased significantly, but was the highest score on the scale (4.1 ± .80).

**Correlation of sample characteristics with pre-posttest changes in satisfaction with use, diabetes knowledge, self-management, and attitudes.** To further understand whether the results of each scale (diabetes knowledge questions, confidence in diabetes self-care behaviors, feasibility and satisfaction with robot use) differed by gender, age group, length of diabetes history, education level, and experience with technology (including: experience with the Internet, experience with mobile applications, and experience with robots), the results of the Mann-Whitney validation analysis were used to understand the differences between the variables on the scales, as shown in Table 5.

Significant differences were found in the application feasibility questionnaire and in the knowledge of diabetes question between those with less than a secondary school education and those with more than a high school education. This effect of education level was evident in the post-test of the diabetes knowledge question. The post-test scores were significantly higher for those with high school education and above than for those with education up to and including secondary school ($z$ = -2.724, $p$ = .006). However, there was no significant difference in the pre-test ($z$ = -1.385, $p$ = .166). In addition, according to the satisfaction scale item "Interacting with the community pharmacy robot facilitates my communication with the community pharmacist." ($z$ = -2.138, $p$ = .033), those who had no experience with using a mobile application had a significantly higher satisfaction score.

## Results of the qualitative interviews with diabetic patients

Five themes emerged from the qualitative interviews with the diabetic patients: Theme 1: Satisfying opportunities for self-directed learning in middle-aged and old age; Theme 2: Reducing wariness and creating comfortable interactions; Theme 3: Vibrant and enriching opportunities for interaction; Theme 4: Robotic applications are not invariably beneficial; and Theme 5: Functional applications are beneficial.

**Table 5. Correlation analysis between the pretest-posttest and participant characteristics [a-d].**

| | Gender | | | Age | | | Education Level | | | Diabetes history | | | Internet experience | | | Mobile application experience | | | Robotics experience | | |
|---|---|---|---|---|---|---|---|---|---|---|---|---|---|---|---|---|---|---|---|---|---|
| | Male (n = 15) | Female (n = 15) | p | 45-64 years old (n = 8) | 65+ years old (n = 22) | p | Below junior high school (n = 13) | High school and above (n = 17) | p | Less than 10 years (n = 15) | More than 10 years (n = 15) | p | N/A (n = 15) | Yes (n = 15) | p | No (n = 13) | Yes (n = 17) | p | No contact experience (n = 12) | Had contact (n = 18) | p |
| **Correlation analysis between satisfaction and participant characteristics** | | | | | | | | | | | | | | | | | | | | | |
| Total score | 8.20 (3.23) | 7.80 (3.28) | .320 | 8.50 (3.46) | 7.82 (3.17) | .229 | 7.23 (3.88) | 8.59 (2.55) | .672 | 6.93 (4.15) | 9.07 (1.28) | .674 | 7.60 (3.72) | 8.40 (2.67) | .113 | 7.15 (3.83) | 8.65 (2.57) | .360 | 6.67 (4.12) | 8.89 (2.11) | .109 |
| **Correlations between pretest-posttest and participant characteristics for application feasibility** | | | | | | | | | | | | | | | | | | | | | |
| pre-test 1 | 3.60 (.83) | 3.27 (1.16) | .596 | 3.50 (.93) | 3.41 (1.50) | 1.000 | 3.38 (1.04) | 3.47 (1.01) | .876 | 3.33 (1.29) | 3.53 (.64) | .947 | 3.47 (.99) | 3.40 (1.06) | .740 | 3.23 (.93) | 3.59 (1.06) | .305 | 3.08 (1.08) | 3.67 (.91) | .087 |
| pre-test 2 | 3.80 (.68) | 3.47 (.99) | .45 | 4.00 (.76) | 3.50 (.86) | .178 | 3.46 (.78) | 3.76 (.903) | .152 | 3.67 (1.11) | 3.60 (.51) | .492 | 3.47 (.64) | 3.80 (1.01) | .137 | 3.46 (.66) | 3.76 (.97) | .196 | 3.50 (.80) | 3.72 (.90) | .293 |
| pre-test 3 | 3.73 (1.03) | 3.20 (1.01) | .12 | 4.00 (.756) | 3.27 (1.08) | .092 | 3.08 (.95) | 3.76 (1.03) | .024* | 3.40 (1.30) | 3.53 (.74) | 1.000 | 3.33 (.976) | 3.6 (1.121) | .405 | 3.23 (.927) | 3.65 (1.115) | .251 | 3.50 (.798) | 3.44 (1.199) | .858 |
| pre-test 4 | 3.73 (1.03) | 3.40 (.99) | .306 | 4.00 (.76) | 3.41 (1.05) | .175 | 3.31 (1.03) | 3.76 (.97) | .164 | 3.47 (1.30) | 3.67 (.62) | 1.000 | 3.47 (.99) | 3.67 (1.05) | .609 | 3.31 (.95) | 3.76 (1.03) | .201 | 3.50 (.80) | 3.61 (1.15) | .440 |
| post-test 1 | 4.20 (.78) | 4.00 (.85) | .545 | 4.25 (1.04) | 4.05 (.72) | .323 | 3.92 (.76) | 4.24 (.83) | .206 | 4.07 (1.03) | 4.13 (.52) | .875 | 4.07 (.80) | 4.13 (.83) | .720 | 3.92 (.76) | 4.24 (.83) | .206 | 3.83 (1.03) | 4.28 (.58) | .226 |
| post-test 2 | 4.20 (.78) | 3.80 (1.21) | .461 | 4.13 (1.36) | 3.95 (.90) | .324 | 3.69 (1.03) | 4.24 (.83) | .091 | 3.87 (1.36) | 4.13 (.52) | .911 | 3.87 (1.06) | 4.13 (.99) | .434 | 3.77 (1.01) | 4.18 (1.02) | .191 | 3.67 (1.37) | 4.22 (.65) | .386 |
| post-test 3 | 4.20 (.78) | 3.93 (1.03) | .545 | 4.13 (1.36) | 4.05 (.72) | .323 | 4.00 (.82) | 4.12 (.99) | .470 | 4.07 (1.16) | 4.07 (.59) | .560 | 4.07 (.80) | 4.07 (1.03) | .720 | 4.00 (.71) | 4.12 (1.05) | .406 | 4.08 (1.17) | 4.06 (.73) | .522 |
| post-test 4 | 4.27 (.70) | 3.67 (1.18) | .147 | 4.13 (1.36) | 3.91 (.87) | .264 | 3.92 (.86) | 4.00 (1.12) | .535 | 3.87 (1.30) | 4.07 (.59) | 1.000 | 3.93 (.84) | 4.00 (1.13) | .539 | 3.77 (.83) | 4.12 (1.11) | .132 | 3.92 (1.24) | 4.00 (.84) | .911 |
| **Correlations between pretest-posttest and participant characteristics of diabetes knowledge** | | | | | | | | | | | | | | | | | | | | | |
| Pre-post Differences | 1.07 (1.49) | 1.33 (1.50) | .579 | .50 (.76) | 1.45 (1.60) | .117 | .92 (1.60) | 1.41 (1.37) | .291 | .93 (1.33) | 1.47 (1.60) | .326 | 1.07 (1.67) | 1.33 (1.29) | .468 | .77 (1.36) | 1.53 (1.51) | .168 | 1.17 (1.59) | 1.22 (1.44) | .711 |
| **Correlations between pretest-posttest and participant characteristics of diabetes self-care behaviors** | | | | | | | | | | | | | | | | | | | | | |
| Pre-post Difference | 1.03 (1.56) | -.21 (1.41) | .072 | 1.22 (1.22) | .11 (1.45) | .320 | -.077 (1.82) | .78 (1.33) | .411 | .33 (2.07) | .49 (.97) | .915 | .04 (1.75) | .78 (1.38) | .577 | -.0769 (1.53) | .77 (1.53) | .745 | .02 (1.95) | .67 (1.30) | .844 |

[a] N(SD)

[b] * p<0.05

[c] Note for question number.

1. Interaction with the community pharmacy robot can help me increase my knowledge of health-related illnesses.

2. The community pharmacy robot can help me with disease management.

3. Community pharmacy robots can help me reduce my anxiety related to my illness.

4. Community pharmacy robots can help facilitate my interactions with community pharmacists.

[d] total score of 20 for feasibility of application (strongly agree = 5, agree = 4, no opinion = 3, disagree = 2, strongly disagree = 1).

**Theme 1: Meeting the opportunities for self-directed learning in middle-aged and elderly individuals.**   Health education was the main function of the robot in this study. The aim was to increase and enhance the knowledge of diabetic patients about diabetes through this interactive process and to make them more aware of how to deal with the disease. Three diabetics in particular said that the information provided by the robot was really helpful to them.

*I feel that this has increased my knowledge. . .about what to eat and what to watch out for. . .*

[Patient 25, Female, 71 years old, with technology experience and an elementary school education]

Since the interaction with the robot is quite straightforward, people don't feel like bothering others or hesitate to worry about what others think, and can decide on one's own what to ask or when to interact, so people can be more active in learning more information through the robot. What the robots can do is not only increase the patient's awareness of their own health, but they can also turn passivity into initiative, allowing people to be more proactive in their learning.

*You'd be embarrassed if someone told you that. . . if you go to the robot like this. . . you can find out for yourself. . . it's not annoying, and you can get information. . . it's acceptable, so I think it's really good. . . no one is bothered. I think it's really good. . . no one's bothering you, you can just take your time and ask questions. . .*

[Patient 03, Female, 50 years old, with technology experience and a middle school education]

**Theme 2: Reducing wariness and creating comfortable interactions.**   Although the robot has a human character and a humanoid image, it is easier for people to let down their guard during the interaction. Diabetics don't think they will be blamed for the incorrectness of their answers, so they can tell the truth with a relaxed attitude.

*It's not like you're under pressure. . . If you talk to a robot, it's okay to be right; it's okay to be wrong, and you won't be punished. . .It's not like a robot is questioning you;*

[Patient 16, Male, 78 years old, with technology experience and a graduate school education]

*people have a personality, they don't like to be told. . .as far as I'm concerned. . .no one will chatter, and no one will get annoyed. . .sometimes you ask someone to ask a doctor and sometimes they can't. . .so I'm telling you, it's better to have something like a computer than to have people telling you that.*

[Patient 03, Female, 50 years old, with technology experience and a middle school education]

**Theme 3: Lively and enriching opportunities for interaction.**   Robots are versatile and can present more than just a single mode of interaction. Three of the six male diabetic patients described the richness of the robot interaction, which not only increased the level of attention directed to the robot, but also made it easier for people to interact with it.

*There may be questionnaires and prizes to be won. . .rewards. . .It's better to say what you want to ask; just look at its expression, and you can probably tell what it wants to ask, so it will attract people's attention. . .and they will focus on the robot.*

[Patient 13, Male, 56 years old, without technology experiences and a high school education]

*The recordings of the robots. . . also have some images and some voices, which are a bit richer and are more acceptable through robots*!

[Patient 16, Male, 78 years old, with technology experience and a graduate school education]

The novelty of the content and the mode of interaction create interest people, they want to know what the robot will say and how it will operate. However, on the other hand, people also say that once the novelty wears off, they don't want to use it anymore. It also means that there has to be new and different content to keep people curious.

*When you ask the same questions two or three times, it's like when you go to health education two or three times, and the health educator says the same thing, you don't want to hear it anymore. . .*

[Patient 19, Male, 58 years old, with technology experience and a university education]

**Theme 4: Robots cannot be invariably used.** The use of robots may seem to be booming, and there are many expectations, but there are still a lot of problems that may exist in terms of promoting the use of robots, starting with the tug of war between technology and age. The majority of respondents in this study reported that older people may have problems using technology. They may not know how to use it, or they may be too afraid or reluctant to try it without experience. If it takes time and effort to learn, this can lead to feelings of rejection and doubts about the need to learn any more. If they are willing to put in the effort to learn, they will need to be guided and taught further. Without leadership, it can be a bit confusing, and it takes time to adapt and understand, which is a problem that must be overcome.

*If someone teaches you, you know how to use it. If you don't tell or teach it, you don't know what it (the robot) is going to do. . .and it (the robot) doesn't know what you are going to do. Now it's all about guidance. . .and the old people don't know how to use it. . .When I don't know how to use a computer, I ask my grandson. . .otherwise I don't know. . .I don't even know how to use that thing in my phone*!

[Patient 25, Female, 71 years old, with technology experience and an elementary school education]

*It's just that older people may not be able to use the new technology, or they may be afraid to use it. They will feel a sense of rejection when they really see or touch it. . .*

[Patient 19, Male, 58 years old, with technology experience and a university education]

While the age gap can be a challenge, it is argued that even so, it is not likely to be a permanent problem. In particular, as technology becomes more embedded in our lives, and the level of education about technology increases significantly, people will become more familiar with

operating technology and living in an environment full of technological stimuli, and they will become more accepting of it.

*In this way, the patient's knowledge may be higher in the future. . . Because after all, the elderly will gradually disappear, and the younger generation will be more and more accepting of this kind of computer stuff and will learn to use it themselves. . .*

[Patient 03, Female, 50 years old, with technology experience and a middle school education]

At the same time, it was felt that there were limitations related to what robots can do. Three diabetic patients mentioned and made it clear that robots cannot do everything and are still limited to certain basic functions, and that the core problems have to be solved by humans.

*The main problem is blood pressure measurement*! *It can't be a doctor either*! *If a robot is a doctor, there will be problems. And the robot can't be responsible*!

[Patient 26, Male, 76 years old, with technology experience and an elementary school education]

When people interact with robots, the communication is also scrutinized and its accuracy is even questioned. Three diabetic patients emphasized that they did not know how to judge the accuracy of the information delivered by the robot. The position of the robot is ambiguous. If it looks like a toy, it may not be suitable for communicating serious matters. People may not necessarily buy into everything the robot has to say.

*It depends on the person. Why should I trust you (the robot) to be accurate*? *Some people have the mentality that. . . Can a robot really be accurate*?

[Patient 03, Female, 50 years old, with technology experience and a middle school education]

*It seems to be just a toy, so it has less credibility.*

[Patient 13, Male, 56 years old, without technology experience and a high school education]

One respondent further commented that compared to human interactions, interactions with robots are less warm, more rigid, and lacking in real emotion, and therefore lack a sense of authenticity. This can be very direct and intense.

*It's better to be able to interact with the staff. . . Sometimes when you're talking to someone, if you don't have that warmth, you feel as if they're not answering your questions to your satisfaction, or you don't have that sense of presence. This is one of the more direct feelings I have.*

[Patient 14, Male, 72 years old, with technology experience and a high school education]

What's more, robots can malfunction as much as any other technology, as if they are carrying a hidden bomb. Two diabetic patients both mentioned what would happen if the robot failed, but they did not share the same viewpoint. One person thought that the human brain would one day be overtaken by the computer, but as long as computer technology is used, there is always the possibility of it malfunctioning, so he would still feel vaguely uneasy. However, one person thinks that the human brain must be better, so he has a more negative view of

the abnormal conditions of robots, and even thinks that when robots fail, they may turn against humans, which will lead to other uncontrollable situations.

*There are times when there is a malfunction, when there is no electricity, and when such problems arise. There are times when problems arise that can harm you. . . Robots are invented by humans too! If it breaks down, won't it destroy you? The human mind is better than anything else. . . Even robots can malfunction! If it malfunctions, the whole process is messed up!*

[Patient 02, Female, 70 years old, without technology experience and a high school education]

*It helps you to solve. . . difficult problems and so on. . . when the computer control program is available. Of course, the human brain can't catch up with the computer, but sometimes the computer has problems too!*

[Patient 25, Female, 71 years old, with technology experience and an elementary school education]

**Theme 5: Each application has its own benefits.** The robot itself has many functions, and there are many different possibilities, but people's imagination and requirements for robots are not all the same, which shows the richness and diversity of the robot's future. This viewpoint is gender-specific. Men tend to talk about how the internal functions of robots can be used, but in the case of women, most focus on the cute appearance of robots, which is one of the main reasons why people want to interact with them.

*I think it's cute, and I want to interact with it, so I think it's cute, and I think it's more approachable. . . so it's probably very well accepted by people.*

[Patient 03, Female, 50 years, with technology experience and a middle school education]

Differences between ages can be seen based on who needs the robot more, and also based on the demand for the robot. For example, when it comes to the functions that a robot should have, older people over 70 years old mainly wanted a robot with a service function, but diabetic patients under 70 years old did not specifically express this need.

*It will bring my tea when I'm lying in bed. . . It will answer you whatever you ask it.*

[Patient 26, Male, 76 years old, with technology experience and an elementary school education]

Of course, differences in personality can also make a difference on their level of need. Personality also affects the willingness of individuals to accept technological interventions such as robots. Some people think that those who are more optimistic and generous will have a higher chance of being receptive to interacting with robots.

*Some people will like robots very much. More open-minded and optimistic people find robots interesting. . .For example, some people are very proud and don't want to listen to robots.*

[Patient 04, Female, 76 years old, with technology experience and a middle school education]

## Results of the qualitative interviews with pharmacists in community pharmacies

The following three themes emerged from the interviews with the community pharmacists: Theme 1: The development of technology must be from the patient's perspective; Theme 2: Creating new services, and Theme 3: The use of robots must be realistic.

**Theme 1: Technology must be developed from the patient's point of view.** During the interviews, both from the perspective of the diabetic patient and the pharmacist in the community pharmacy, meeting the fundamental needs of patients is the first priority in order to manage the disease well. This is why monitoring, reminding, and recording are so important. Robots can help optimize and implement these basic needs.

The first requirement monitoring physiological data, such as blood glucose or general physical data, and half of the pharmacists stressed the importance and necessity of monitoring. The two pharmacists mentioned that the monitoring function should be instantaneous, but to achieve this, more electronic technology products are needed, such as electronic bracelets.

> *A bracelet can be designed to be worn by the elderly. The robot can automatically monitor pulse rate and blood oxygen, and can quickly notify the authorities and send a message to the family.*

[Pharmacist 01, Male, 28 years old, 1 year of work experience]

In terms of medication-related issues, more than half of all of the pharmacists were concerned about patients taking their medication correctly and on time. The most practical and straightforward way to prevent patients from forgetting to take their medication, and to prevent any left-over medication, is to remind them of it, and it is not only practical but also basic and important.

> *I think the first thing they need is a regular reminder to take their medication. . .because taking medication, although important, is easy to ignore, so they need to be reminded. . .*

[Pharmacist 01, Male, 28 years, 1 year of work experience]

Furthermore, disease management should not only be about the robot receiving information from the patient, but also about the patient receiving the overall interactive content, so that the overall interaction is more positive. The main focus of robotic applications is to interact with people in order to provide effective assistance, so the way in which information is transmitted is also important.

> *In the case of paper, we could consider. . . printing them out. . . That way at least they have something more tangible, they can see the reminders or the data from today's measurements, and that's something that's quite practical.*

[Pharmacist 08, M, 44 years old with 10 years of experience]

Almost all pharmacists agree that personalization of messages should be enhanced. Regardless of the differences in the patient's background knowledge and the length of their medical history, it is important to pay attention to what information the patient really needs. This is an important core concept in health education, and it is important to emphasize and apply this concept with the help of technology in order to truly assist patients in their disease management.

*Patients with different stages of diabetes may have different health care needs. . . For example. A diabetic with a primary diagnosis might have. . . It's just that different people have different needs. . . Customization. It's not the same for everyone, either. That is, what you provide is not necessarily what the person needs.*

[Pharmacist 03, Male, 35 years, 5 work years]

We are not looking after diabetes, we are looking after the patient with diabetes. This means that all issues need to be taken into account. When any one factor is overlooked, care for the disease is not as complete as it could be. The three pharmacists explained that, in general, diabetic patients may have more than just a single disease. Most commonly, they have a combination of hypertension and hyperlipidaemia, and the diseases often interact with each other. Therefore, a more holistic approach should be taken when considering a patient's problem.

*Because diabetes is a chronic disease, people with diabetes usually don't have only one blood sugar problem. If the treatment only focuses on diabetes and its complications, this focus may be a bit narrower.*

[Pharmacist 02, Male, 36 years old with 8 years of work experience]

From the patient's point of view, there are health issues that should not be ignored. There can be many precursors to complications of diabetes, but whether the patient is aware of them is a major issue. When a patient is faced with an abnormal condition today, do they know the right way to deal with it? Is it something that is selectively ignored or not considered very important? It is often difficult to distinguish between a problem of ageing and an abnormality caused by a disease. When small problems are ignored and become major illnesses, they can easily lead to difficulties in follow-up care.

*Because a lot of people, especially the older generation, are not as alert to this kind of health problem. They think it's okay, so when they are willing to go to the doctor, it's usually when it's more serious. . .They may think they can just go to bed and wake up, but it's actually a sign of a problem. . .*

[Pharmacist 08, M, 44 years old with 10 years of work experience]

**Theme 2: Creating new services.** Patients may be trapped in a human-to-human communication framework. Some problems may have been there all along, but there was not an opportune time to intervene. For community pharmacists, robotic assistance may also make a difference to the pharmacy. Firstly, eliciting conversations can be a powerful tool for community pharmacists. Three pharmacists each provided similar insights. Patients are afraid to raise issues that they avoid talking about, or that they think may be unnecessary, or that they are embarrassed or ashamed of. If the robot can draw the patient's attention to the issue and make the patient more passive, the pharmacist will have the opportunity to intervene further. Therefore, the robot has the opportunity to play a key role as an intermediary to initiate the conversation.

*It doesn't make people feel pressured or constrained. Perhaps if they were to communicate the same information in a different way, they would be more receptive and more willing to follow such a suggestion. . .Because if a person talks to a person directly like that, you might think,*

*"How can you talk to me like that? But maybe a robot could play a role in communicating health information directly, so that people would notice. . .It (the robot) could take the lead.*

[Pharmacist 08, M, 44 years old with 10 years of work experience]

Another advantage of robots is that they can provide a richer variety of resources. Since the robot has a sound system, movement, video images, and animated videos, it can be used to present precise images or audio-visual interactions in a more immediate manner. This means that pharmacists can have a vast database from which they can extract resources at any time, making it possible to deliver health education messages to patients in a powerful way. Pharmacists should make good use of such a tool, so the robot can assist them with presenting health information in a realistic way, so that the patient can have a concrete and profound understanding, and so that health education can be conducted more smoothly.

*Because there will be sound, and then there will be pictures, you can actually get the patient into the situation right away. It's not like when we're doing health education here. If I don't have pictures, if I suddenly go and bring him a mango, I can't do anything about it. . . That's the advantage of it, because you can actually photograph it. . . It's better than a single picture because you can actually take a picture. . . there's a physical picture, and from the robot, I can order a lot of pictures to show the patient. . . It's about making it very tangible.*

[Pharmacist 10, Female, 60 years old with 35 years of work experience]

By using the robot to provide specific information, the pharmacist can use the interaction between the patient and the robot to understand the patient's deficiencies or misconceptions, and then intervene appropriately to improve the specific content. Therefore, it is important that the robot and the medical practitioner work together so that the patient can benefit from both.

*If humans and robots work together, this will be more clearly reflected in the health education. . .After patients have seen the health information, they can ask educated professionals if there are any problems. If they have any questions, they can ask them immediately. . .*

[Pharmacist 01, Male, 28 years old with 1 year of work experience]

Although we always talk about robots enhancing certain functions and providing more services, this is only true for a robots working alone. In fact, there should be a focus on building stronger networks. There are two groups of people interacting with the robots, patients, and pharmacists, so there should be a more systematic link among the three. There is a need to document and systematize information, and to do so in a meaningful way that provides feedback to patients. Only then can there be effective two-way communication. Two-way, complete feedback is an integral part of a robotic system. Therefore, robots help connect everyone to the institution and create a more systematic, complete healthcare network.

*It takes a picture of you, your height and weight, or your blood sugar. You can take a picture of your weight, your height, or your blood sugar, and it'll record it for you. . .This can be uploaded to LINE or the cloud, or can become your medical history. Then, when you go to the doctor, the doctor will see that you went to the XX pharmacy on a few months and days and interacted with the robot, and the doctor will know what your condition was at that time..*

[Pharmacist 07, Male, 59 years old with 30 years of work experience]

**Theme 3: Robotic applications must be realistic.** Advances in technology have created a lot of possibilities for robots, and people do expect this. But in retrospect, we have to be realistic in our response to technological development. If a robot does not have the flexibility to interact with a stable system, it will not be able to perform a task on its own. As a result, there may be many problems with their practical use. Robots are supposed to assist community pharmacists However, when the robot is unstable, and the pharmacist is required to spend extra time fixing the robot, the true purpose of the application is lost. Instead of being truly effective, it may have other negative effects. It is one thing for the robot and the pharmacist to work in tandem, even if the robot is only playing a supporting role, but when they work together, it should go smoothly.

*If you want it to really be able to stand alone, it can't get stuck. . .It's like asking me to help it. . .if it's busy in the pharmacy, it needs to be on its own! It needs to be able to work alone. . .it needs to be fluid and agile. So it's about having a little helper or something like that around that is able to work independently/alone.*

[Pharmacist 06, Male, 44 years old with 19 years of work experience]

Another thing to consider is the actual purpose of the robot's intervention. On the one hand, it is a question of what exactly is intended for the patient. The current prototype robot is not innovative enough and may provide the advantage of combining technology and novelty. However, each intervention, each interaction, must have a meaning that is intended to be conveyed. It is therefore important to think carefully about how the overall intervention will really make a difference to the patient and how the patient will feel about it. If the patient is not considered to have a memorable point, then the intervention becomes rather weak.

*It's very much like a child playing a game. . .it's less meaningful. . .It has a function, but the added value is a bit better. . . or the basic steps of a health check. . .Not necessarily blood glucose, but at least height and weight, so that the first time user will be impressed or not get nothing out of it. . .*

[Pharmacist 07, Male, 59 years old with 30 years of work experience]

At the same time, the role of the robot was discussed in terms of how it was defined. As we discussed earlier, robots can be used to improve the quality of health education. However, if the current system is already complete, why would we need robots to make further breakthroughs?

*For example, we now have health education teachers who give very informative lectures and take into account every aspect, including nutrition, exercise, and how to inject insulin. In fact, the health education teachers are very good in every way.*

[Pharmacist 09, Male, 60 years old with 30 years of work experience]

## Discussion

A total of 30 diabetic patients interacted with the prototype robot in the study. Based on the results of the interaction, it was found that the interaction with the robot could increase the

knowledge of the diabetic patients about diabetes. The diabetic patients felt that it was feasible for the pharmacy to integrate the robot. The diabetics were generally highly satisfied with the process, content, social orientation, and information quality of the robot. Also, 12 diabetic patients and 10 pharmacists from community pharmacies were interviewed. Based on the findings from the interviews, for diabetic patients, robots can bring a richer and more diverse range of interactions. Such communication not only has the potential to facilitate independent learning, but can even reduce patient stress on some levels. In terms of robot choices, it is not possible to satisfy everyone's needs at the same time, as people have different views on robots. Even though robots can bring many benefits, we see the potential dilemma when considering them as an intervention. For pharmacists in community pharmacies, robots can indeed provide an advantage in complementing health care or community pharmacies, and can even create new services. At the same time, however, diabetes care must not be too narrowly focused on the patients' underlying needs, but must be comprehensive. In terms of practical applications, it is also expected that the future development of robots will face many challenges.

One interesting finding of this study was that conversations with the robot were considered less stressful than conversations with a human. Diabetic patients are worried about being blamed for what they say during human dialogue. However, when interacting with a robot, diabetic patients are indirectly relieved of psychological stress. Previous studies have used two different interviewers, a human and a robot, to interview respondents about sensitive issues. However, the results showed that humans were still better at interpreting messages, and the respondents felt that they were still better understood when talking to a human [22]. Therefore, while we can take advantage of the fact that robots do not cause additional stress to patients, we need to be careful about whether they can clearly understand the issues that patients are communicating. There may be an infinite scope for robotics, but what humans can do, robots are not necessarily suited to do. As one diabetic patient in this study said, robots cannot do everything, and they have to rely on humans for many things, and interaction with a human is more real and warmer. The results are also in line with previous studies, which emphasize that human interaction cannot be replaced by robots [23]; in practice, people prefer human contact when communicating [24]. However, the services and information that robots can provide are recognized. For example, older people can learn something new, get information, or be assisted with instrumental activities of daily living (IADLs) [24] through robots. Some people may even think that robots are sometimes better than humans for taking blood pressure measurements, for example, because they are more accurate [25]. As in this study, the diabetic patients felt that the robot was suitable for blood pressure measurement but not for diagnosis, on the grounds that the robot could not take responsibility for this.

There was no consistency in the views of the middle-aged and elderly diabetic patients on the appearance, functionality, and suitability of robots, which suggests that robots are evolving for middle-aged and elderly people, and that it is not always possible to find a fixed model to suit everyone. In terms of appearance, which is the first impression of a robot, women in this study specifically expressed a preference for cute robots, which they found more attractive. It is true that a cute robot makes people more comfortable with close contact, use, and trying to understand the robot, which also suggests that the appearance of the robot affects the degree of user acceptance [26]. However, the middle-aged and elderly men in this study were not particular about appearance and did not express negative opinions about the appearance of the prototype robots, but rather discussed the functionality of the robots. Interestingly, in previous studies, men were even mentioned as not being particularly fond of cute robots in terms of their appearance, probably due to their concern for what others might think [27]. Only one respondent in this study said that he liked humanoid robots and thought that they were sophisticated enough. However, in fact, the opinion about humanoid robots is quite polarized.

Some even think that it is weird to have a robot act like a human being when they are not one [26]. In addition, the size of the robot is also a feature that the older people focused on since it is important to how they interacted with the robot or the applicability of the robot. While most diabetic patients said that the robot should be larger and easier to interact with, others felt that it depended on the main function of the robot or its suitability in the field. This is in line with previous literature. There does not seem to be a definitive conclusion on the best size for a robot. That is, while most elderly people said that a robot of at least 100–120 cm would be more suitable, some argued that a smaller robot of 100 cm would be sufficient [28].

This study showed gender differences in appearance preferences, while the choice of function was influenced by age. In the study, older people at or over the age of 70 were also more likely to be able to provide substantive services (e.g. functions such as pouring tea), but younger people did not make a similar point. Past studies have also mentioned functions that people in their seventies and eighties would like a robot to provide, such as helping them to get up and carrying heavy objects [28]. Perhaps the services mentioned above are more direct services to the elderly than information or entertainment. In particular, robots can provide real relief to elderly people when they have mobility problems and need direct help.

In the study, we were able to determine that apart from community pharmacies, the diabetic patients felt that robots could be developed in areas such as community centers, institutions, hospitals, clinics, and at home. In-home care robots are one of the main focuses of robotics development today. These robots can be used to monitor abnormal conditions in the elderly, such as falls, life monitoring, or environmental assessments, as well as other safety concerns. In addition, they can also be used to assist with activities of daily life and remind the elderly of their affairs, making it a good companion for them at home [29–32]. This is similar to the expectations of the diabetic patients and community pharmacists in this study. In addition, although diabetic patients have indicated that hospitals can also be a place to promote these types of health robots, at present, the robots in hospitals are still mostly functional in nature, such as assisting with surgical operations, fully automated instrumentation, transport and delivery, and so on [33–35]. Few studies have placed robots in hospitals in a socially functional way. At the same time, not all patients agree that robots are suitable for use in hospitals. In an adequately resourced hospital environment, robots need to be clearly positioned so that they do not appear superfluous. As for the use of robots in organizations, they are currently mostly used for companionship and cognitive and social functions [9]. It is clear that the functions that a robot can perform are very different for different settings and users. It may be possible to try to attempt to apply them to different contexts, but it is also important to consider what is of most concern to the user in such contexts and whether robot interventions are meaningful to that user.

The application of robots should not be limited to stand-alone operation, as this would limit developmental possibilities. Many robot applications combine robots with blood pressure meters, weight scales, and muscle meters to provide multi-dimensional care through the Internet [36]. They also combine social robots with smartphones and smart bracelets to implement a more systematic, integrated approach to disease management [37]. In addition, linking or providing feedback to patients is also important, and a more robust system is needed to turn one-way interactions into effective two-way interactions. As emphasized in this study, further evaluation of the data collected by the robots requires back-end expertise. This will not only help pharmacists understand the patient better and make it possible to intervene in a timely manner in a community pharmacy, but information linkage will also allow the doctor to have a better understanding of the patient's health status at return visits. In the past, there have been references to feeding information about children into the medical system and giving advice from a distance through medical professionals. Therefore, not just the social robots but entire

systems that can provide correct information in its entirety are needed [37]. Therefore, in addition to the functions provided by the robots themselves, it is also necessary to take into account the establishment of front-end and back-end systems for the application of robots in healthcare. This is because healthcare is a multi-disciplinary, multi-directional service that requires a full range of services, and this is the main focus of robot development.

The active development of robots is clear, but this study also shows that the attitudes of older people towards robots are not always positive. Most diabetic patients point to the difficulties that older people may have with using technology. This is also in line with the literature suggesting that age differences can lead to different attitudes towards robots in terms of acceptance [38]. In fact, it is not necessarily age itself that affects attitudes towards robots, but the fact that one has never used a robot before that makes use unlikely or causes people to be afraid to use one. Indeed, attitudes come mainly from personal experience with using technology or robots [24]. A lack of experience can easily lead to resistance to the use of robots and a feeling of insecurity about such use [39]. Literature from Taiwan also shows that older people find it difficult to use some tablet technologies and panic when they don't know how to fix a problem. When errors occur, older people may also be confused as to how to correct them [40]. However, there is also an important point to be made in this study, which is that these problems, while they exist, are not insurmountable. The more information and education people receive, the more easily such problems will be solved. However, some problems are not so easy to solve. Older people may be concerned about robot malfunctions and may even be distrustful and reject them as a result. Instability in robots affects people's perception of the use of technology [41]. In a previous study on the use of robots to manage chronic obstructive pulmonary disease (COPD), when the robots were placed in the home, the overall quality of the intervention was affected by malfunctioning of the robot [10]. Therefore, the malfunctioning of robots implies system instability, and the fact that it is not easy to troubleshoot technology products, for example, adds to the level of unease and distrust in the use of technology. Other commonly cited negative thoughts about robots include privacy issues. In particular, when robots are used in the home environment [42], sometimes strong words such as 'invasive' are used to describe the experience, and can be difficult to perceive a robot as being able to provide real help [38]. The main reason why this was not mentioned in our study is because the interaction in this study occurred in an open space, so there was no sense of being under surveillance at home, and use in the community is less of a concern. The common emphasis, both at home and in the community, is on whether there will be procedural problems with robots, which is of great concern to everyone. When technology is accepted by older people, then a robot intervention will be meaningful [43]. Understanding and eliminating the dilemmas of technological interventions will be the only way to truly benefit older users.

There are many different applications of robots for the elderly, but there is a relative lack of applications for elderly diabetes care. Currently, robots are mainly used in elderly health care to provide direct physical assistance, companionship, or health and safety monitoring [44]. The direction and future goal of the prototype robot in this study is to focus on the seven indicators of diabetes care, AADE7 [45] and to build a complete care model as a result. In the past, the use of technology to promote diabetes care has mainly focused on healthy eating, blood glucose monitoring, and medication [46]. In another paper, it was found that more than 70% of mobile devices have a healthy diet and monitoring function, and more than half have a medication function. In spite of this more than half having a medication function, just under 30% have risk reduction, health adaptation, and problem solving, and only around 5% provide additional knowledge about diabetes [47]. The authors go on to explain that it is relatively easy to present functions for diet, medication, exercise, and monitoring because they can be clearly recorded, and reminders can be given. However, when it comes to problem solving, risk

reduction, etc., the definition of such functions is very vague. In addition, with today's information explosion, people can easily access a large amount of information they want on the Internet or in the media, which is not only fast but also relatively informative, but this also makes it difficult for technology applications to focus on this. Technology may not always provide the best assistance in terms of emotional adjustment and social support since such adjustments are a psychological issue [47]. During the development of the prototype, it was intended that all seven functions would be represented in relation to the seven care indicators, but the weighting of the seven care indicators could not be evenly distributed in the overall performance to match the most appropriate and feasible interaction model. Overall, the content of the health education topics will determine the type of information that patients can access. However, today's information sources, while vast, are littered with fake news, commercially focused interests, or questionable information that has yet to be verified. It can be challenging for users, even for older people, to filter, discern, and assimilate the correct and necessary information from these sources. Therefore, it may help people to obtain information that is more useful to them if it is filtered, and scrutinized, and accuracy is ensured.

In the area of disease management, it seems easy to overlook the fact that when we develop technological interventions for diabetes, we tend to focus on diabetes and ignore patient care. It is also a common point made by pharmacists and diabetic patients that a diabetic patient does not usually have just one disease. When diabetic patients have more than one co-morbidity, their needs may also include how to manage multiple diseases at the same time. In previous studies, although there may be opportunities for technological interventions that cover multiple diseases, they have not yet been developed in any depth, mainly in terms of physiological data or records, for example: positive or negative feedback on changes in physiological data in patients with both diabetes and hypertension [48], teaching videos on blood pressure or blood glucose [49], and co-management of a disease through data recording or monitoring [48, 49]. In a study discussing hypertension, it was also highlighted that blood pressure can be managed through mHealth. However, hypertension, which is also related to cardiovascular disease, is a less developed part of technology management at present [50]. Therefore, in disease management, when one disease meets another, it is not just one plus one. We need to think holistically about the most appropriate recommendations, treatment goals and priorities for the patient at the time.

## Strengths and limitations

**Strengths.** The strengths of this study are threefold: The first is that the robot is placed in a real-world setting rather than just in a perfectly controlled laboratory. If the robots were only in a laboratory, we could indeed sense the functionality of the robots and experience the interaction with them without interference. However, the real application of robots requires a real connection with people and the environment. Through this study, it is possible to better understand the problems that need to overcome when robots are affected by the external environment. Also, when there are other influences on the interaction between humans and robots, lessons can be learned on how to adapt robots for future applications in real-world environments. The second is the integration of robots and the community pharmacists in the study. In the past, few studies have directly aligned robots with medical staff. Robots have the opportunity to provide care for diabetes, but more complete care for the disease requires more intensive professional involvement. With this prototype robot, we explored the possibilities of collaboration between robots and professionals. By breaking away from the stand-alone model of robotics and linking with community pharmacists, the robot not only can be used to a great advantage, but also can help improve the quality of care provided by community pharmacists.

The third strength focuses on capturing real feedback from community pharmacists and diabetic patients to understand the real needs from both the medical and patient sides, so that a diabetes care robot can be developed and applied in a user-centered manner in the future. The prototype was initially built based on the literature and professional and team discussions, so that users could have a realistic model to experience and then gain a deeper understanding of user insights and needs. The application of medical technology should not only focus on effectiveness, but also on the real experiences of the user, in order to have the opportunity to make medical technology interventions not only effective but also sustainable.

**Limitations.** The research limitations of this study are divided into the following points: Firstly, the sample size was small, and the target population was limited. In this study, there were only 30 diabetic patients who participated in the pretest-posttest questionnaire, which was a small sample size. In addition, the area of enrollment was limited to the Tainan City region of Taiwan, which is mostly urban. As a result, it was not possible to explore the use and perception of robots among older people from different cultures and regions, and due to the small sample size, it was difficult to extrapolate the results of the study. Secondly, the single intervention may have led to evaluation biases. This time, the prototype robot was used in a single intervention, taking into account the characteristics of the site and the robot. Since there was no long-term observation of the intervention, the first time it was used, the participants were likely to rate the robot intervention highly positively in terms of novelty and fun. The third limitation is the participants' attitudes. The overall results of the study showed a high level of acceptance of the robots, but given the need to explain the whole intervention process to the participants at the time of case intake, it is likely that if they were not interested in technology, and they would not have been willing to participate in the study. Therefore, the participants in this study were receptive to technology and had high expectations and ideas, which may have led to a positive recommendation. The fourth is the limitations related to the robot. When robots are involved, they need to be warmed up and occasionally encounter program updates, which can cause problems with interaction with participants due to program barriers or the Internet, affecting the patient experience or preventing them from concentrating on understanding the operation behind the technology, thus making them feel uneasy about unstable or unfamiliar programs. Fifth, there was an impact from the case intake environment. In order to get to know the real environment, we took cases in community pharmacies. Although it is always preferable to choose a community pharmacy with the right space and size to receive cases in order to get a more realistic picture of the real-world environment, the large number of people coming and going from a community pharmacy and the presence of business and delivery people in addition to customers, made it easy for participants and pharmacists to be disturbed during the research process, and it was also difficult to control the presence of patients on site other than the participants. During busy periods, the study was more likely to be interrupted and could thus not be implemented smoothly.

## Conclusion

In conclusion, a prototype robot was used as a first step to gain a more realistic understanding of the feasibility and initial results of the application of the robot to community health care for diabetic patients and pharmacists in community pharmacies. Although it was not possible to explore in a single intervention whether robots can have a profound impact on the quality of care in the community and enhance the health care of diabetic patients, the overall direction of the development was recognized by the middle-aged and elderly diabetics in the community. At the same time, community pharmacists also saw the application of technology to community care as forward-looking and feasible, which suggests a potential for the use of robots in

community care. However, considering the realities of the environment, factors such as robot failure, inability to operate independently, or interference from the external environment can make it difficult to operate them in practice. However, we are already making bold and forward-looking attempts in this area in terms of linking the medical side, the patient side, and the technology side. However, disease care requires long-term interventions.

Future research should therefore focus on a larger scale to see the actual effect and longer-term interventions on establishing an effective model between medicine and technology, and on how this benefit can be passed on to patients. It is also important to consider whether such high-tech, high-cost applications can meet realistic cost requirements. The rapid development of technology and its more effective use will bring about positive changes in society.

## Author Contributions

**Conceptualization:** Ching-Ju Chiu.

**Data curation:** Ching-Ju Chiu, Lin-Chun Hua.

**Formal analysis:** Ching-Ju Chiu, Lin-Chun Hua.

**Resources:** Ching-Ju Chiu.

**Supervision:** Ching-Ju Chiu.

**Writing – original draft:** Ching-Ju Chiu, Lin-Chun Hua.

**Writing – review & editing:** Ching-Ju Chiu, Lin-Chun Hua, Chieh-Ying Chou, Jung-Hsien Chiang.

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
