## [Decision Letter · Decision Letter 0]

9 Dec 2021

PONE-D-21-35505Robot-enhanced diabetes care for middle-aged and older adults living with diabetes in the community: a mixed-method evaluationPLOS ONE

Dear Dr. Hua,

Thank you for submitting your manuscript to PLOS ONE. After careful consideration, we feel that it has merit but does not fully meet PLOS ONE’s publication criteria as it currently stands. Therefore, we invite you to submit a revised version of the manuscript that addresses the points raised during the review process.

ACADEMIC EDITOR: Dear Authors, both the reviewers has given positive remark for the manuscript, however, they have suggested minor changes, kindly incoorporate the changes and resubmit.==============================

We look forward to receiving your revised manuscript.

Kind regards,

Rohit Ravi, Ph.D.

Academic Editor

PLOS ONE

Journal Requirements:

Reviewers' comments:

Reviewer's Responses to Questions

**Comments to the Author**

1. Is the manuscript technically sound, and do the data support the conclusions?

Reviewer #1: Yes

Reviewer #2: Yes

2. Has the statistical analysis been performed appropriately and rigorously? 

Reviewer #1: Yes

Reviewer #2: Yes

3. Have the authors made all data underlying the findings in their manuscript fully available?

Reviewer #1: Yes

Reviewer #2: Yes

4. Is the manuscript presented in an intelligible fashion and written in standard English?

Reviewer #1: Yes

Reviewer #2: Yes

5. Review Comments to the Author

Reviewer #1: Firstly, I would like to thank the editor for letting me review this manuscript. It was an interesting one and would congratulate the authors for thinking in the direction of spreading robotic-based diabetes awareness for the elderly and middle-aged people. It must be a difficult task, planning, setting-up, and implementing the study.

As they have highlighted using the robotic approach significantly improved diabetes-related knowledge amongst the patients with diabetes, such studies on a larger scale must be conducted to see the actual effect. Although it's difficult but with collaboration and funding other regions and people with different socio-economic backgrounds could be included.

The data is well-presented and analyses also seems to be extensively done. Although the authors have mentioned in the limitations about small sample size, I think they can consider adding it in the "Methods" section or "Title" perhaps.

In the "Measures" section...please cross-check if there is repetition...Line 165 and line 168.

Reviewer #2: Great effort, congratulations to authors on this detailed manuscript. I don't have any significant comments, but since I am a physician, I find it very tiring text to read and I recommend to shorten the manuscript much further to allow for larger audience of readers to benefit from this paper.

I also would like to ask to include information in the abstract about the population studied in this paper who are middle age and older population. As I read the abstract first, I didn't release you were targeting primary a specific age group and it would be optimal if that information is included in the abstract.

6. PLOS authors have the option to publish the peer review history of their article (what does this mean?). If published, this will include your full peer review and any attached files.

Reviewer #1: **Yes: **Dr. Samreen Siddiqui

Reviewer #2: No

---

## [Author Response · Author response to Decision Letter 0]

12 Jan 2022

Reviewer # 1:

1. Firstly, I would like to thank the editor for letting me review this manuscript. It was an interesting one and would congratulate the authors for thinking in the direction of spreading robotic-based diabetes awareness for the elderly and middle-aged people. It must be a difficult task, planning, setting-up, and implementing the study. As they have highlighted using the robotic approach significantly improved diabetes-related knowledge amongst the patients with diabetes, such studies on a larger scale must be conducted to see the actual effect. Although it's difficult but with collaboration and funding other regions and people with different socio-economic backgrounds could be included.

AUTHOR’S RESPONSE: The authors appreciate the reviewer’s positive evaluation of our work. Due to the instability of the prototype robot, it is difficult to carry out a larger scale research currently. But we added this important point to the conclusion as expectations in the future research (Line 1004, page 63).

2. The data is well-presented and analyses also seems to be extensively done. Although the authors have mentioned in the limitations about small sample size, I think they can consider adding it in the "Methods" section or "Title" perhaps.

AUTHOR’S RESPONSE: The authors appreciate this important advice. A note on this will be added to the methods (Line 116 and line 117, page 7). The sentences were marked with blue. Thank you for the title suggested. The precedent version of the title has been replaced, becoming “Robot-enhanced diabetes care for middle-aged and older adults living with diabetes in the community: a small sample size mixed-method evaluation”.

3. In the "Measures" section...please cross-check if there is repetition...Line 165 and line 168.

AUTHOR’S RESPONSE: The authors appreciate the reviewer for pointing out this problem. Thank you for the detailed review. We have removed the repetitive sentences and marked the removed ones with yellow. 

Reviewer # 2: 

1. Great effort, congratulations to authors on this detailed manuscript. I don't have any significant comments, but since I am a physician, I find it very tiring text to read and I recommend to shorten the manuscript much further to allow for larger audience of readers to benefit from this paper.

AUTHOR’S RESPONSE: The authors appreciate this important suggestion. We have significantly shortened our manuscript from 67 to 62 pages.

2. I also would like to ask to include information in the abstract about the population studied in this paper who are middle age and older population. As I read the abstract first, I didn't release you were targeting primary a specific age group and it would be optimal if that information is included in the abstract.

AUTHOR’S RESPONSE: The authors appreciate this important advice. We have added the age of the diabetes patients in the abstract.

---

## [Editor Report · Decision Letter 1]

2 Mar 2022

Robot-enhanced diabetes care for middle-aged and older adults living with diabetes in the community: a small sample size mixed-method evaluation

PONE-D-21-35505R1

Dear Dr. Hua,

We’re pleased to inform you that your manuscript has been judged scientifically suitable for publication and will be formally accepted for publication once it meets all outstanding technical requirements.

Kind regards,

Rohit Ravi, Ph.D.

Academic Editor

PLOS ONE

Additional Editor Comments (optional):

Dear authors the changes are satisfactory. I appreciate for incoorporating the changes suggested by both the reviewers.
---

## [Editor Report · Acceptance letter]

7 Apr 2022

PONE-D-21-35505R1 

Robot-enhanced diabetes care for middle-aged and older adults living with diabetes in the community: a small sample size mixed-method evaluation 

Dear Dr. Hua:

I'm pleased to inform you that your manuscript has been deemed suitable for publication in PLOS ONE. Congratulations! Your manuscript is now with our production department. 

Kind regards, 

on behalf of

Dr. Rohit Ravi 

Academic Editor

PLOS ONE